# The Association between Prostate-Specific Antigen Velocity (PSAV), Value and Acceleration, and of the Free PSA/Total PSA Index or Ratio, with Prostate Conditions

**DOI:** 10.3390/jcm9113400

**Published:** 2020-10-23

**Authors:** María-Carmen Flores-Fraile, Bárbara Yolanda Padilla-Fernández, Sebastián Valverde-Martínez, Magaly Marquez-Sanchez, María-Begoña García-Cenador, María-Fernanda Lorenzo-Gómez, Javier Flores-Fraile

**Affiliations:** 1Department of Surgery, University of Salamanca, 37007 Salamanca, Spain; maria.flores.fraile@usal.es (M.-C.F.-F.); sebasv_2000@hotmail.com (S.V.-M.); mbgc@usal.es (M.-B.G.-C.); mflorenzogo@yahoo.es (M.-F.L.-G.); 2Section of Urology, Departamento de Cirugía, Universidad de La Laguna, La Laguna, 38071 Tenerife, Spain; padillaf83@hotmail.com; 3Multidisciplinary Renal Research Group of the Institute for Biomedical Research of Salamanca (IBSAL), 37007 Salamanca, Spain; magalymarquez77@gmail.com; 4Department of Urology of University Hospital of Avila, 05004 Ávila, Spain; 5Department of Urology of University Hospital of Salamanca, 37007 Salamanca, Spain

**Keywords:** PSA velocity, free PSA/total PSA ratio, benign prostatic hyperplasia, PIN, prostatitis, prostate cancer

## Abstract

Introduction: Prostate-specific antigen velocity (PSAV) is used to monitor men with clinical suspicion of prostate cancer (PCa), with a normal cut-off point of 0.3–0.5 ng/mL/year. The aim of the study is to establish the predictive capacity of PSAV (value and acceleration) and of the free PSA/total PSA index or ratio. Method: Prospective multicentre observational study in 2035 men of over 47 years of age. Inclusion criteria: men who wished to be informed on the health of their prostate. Exclusion criteria: men with a previously diagnosed prostate condition. Groups: GA: (*n* = 518): men with serum PSA equal to or greater than 2.01 ng/mL. GB: (*n* = 775): men with serum PSA greater than or equal to 0.78 ng/mL and less than 2.01 ng/mL. GC: (*n* = 742): men with serum PSA less than 0.78 ng/mL. Variables: prostate-specific antigen (PSA); age; body mass index (BMI); PSA velocity (PSAV) (ng/mL per year); free PSA/total PSA index (iPSA); PSAV acceleration (increasing: positive, or decreasing: negative); prostate diagnosis (benign prostatic hyperplasia (BPH), prostatic intraepithelial neoplasia (PIN), or infectious and non-infectious prostatitis and prostatic adenocarcinoma (PCa)); de novo diagnoses of urinary tract diseases or conditions; concomitant treatments, diseases and conditions; final diagnosis of prostate health. Results: Mean age 62.35 years (SD 8.12), median 61 (47–94); age was lowest in GC. Mean BMI was 27.89 kg/m^2^ (SD 3.96), median 27.58 (18.56–57.13); no differences between groups. Mean PSAV was 0.69, SD 2.16, median 0.13 (0.001–34.46); PSAV was lowest in GC. Mean iPSA was 27.39 u/L (SD 14.25), median 24.29 (3.7–115); iPSA was lowest in GA. PSAV had more positive acceleration in GA and more negative acceleration in GC. There were 1600 (78.62%) cases of normal prostate or BPH, 322 (15.82%) cases of PIN or non-infectious prostatitis, and 113 (5.55%) cases of PCa. There were more cases of BPH in GC and more cases of PIN or prostatitis and cancer in GA (*p* = 0.00001). De novo diagnoses: 15 cases of urinary incontinence (UI), 16 discomfort/pain in LUT, 112 cases of voiding disorders, 12 urethral strictures, 19 hematuria, 51 cystitis, 3 pyelonephritis, 4 pelvic inflammatory disease; no differences were found between groups. In the multivariate analysis, PSAV and the direction of PSAV acceleration (positive or negative) were the variables which were correlated most strongly with prostate health. iPSA was associated with the presence of prostatitis, PCa, and BPH. Men in GA had more prostatitis, PCa, treatment with alpha blockers, and history of previous smoking. GB had more cases of BPH and more positive acceleration of PSAV. GC had more normal prostates, more BPH, more use of ranitidine, and more PSAV with negative acceleration. Conclusions: PSAV, direction of PSAV acceleration, and iPSA in PSA cut-off points of 0.78 ng/mL and 2.01 ng/mL in a priori healthy men over 47 predict the probability of benign or malignant pathology of the prostate.

## 1. Introduction

Prostate health problems, both benign and malignant, control more than half of the entire male population [1,2]. Although the main laboratory tool for its study is PSA, this is a non-specific and not very sensitive marker, therefore, we propose a study that provides information about the analysis of the velocity and direction of PSA velocity in the male population of the community in its relation to prostate health, both benign and malignant.

Every year, 1.1 million new cases of prostate cancer (prostate carcinoma, PCa) are diagnosed around the world, and 307,000 die from the disease. An estimated 1 in 7 men will be diagnosed with PCa during their lives [3].

The use of prostate-specific antigen (PSA) as a serum marker has revolutionised PCa diagnostics [4]. Prostate-specific antigen, though a marker for the organ, is not specific to cancer; PSA may therefore be elevated in benign prostatic hyperplasia (BPH), prostatitis and other non-malign conditions. PSA and digital rectal examination (DRE) are fundamental tools in the diagnosis of suspected PCa [5]. High PSA levels are associated with suspected PCa, but PSA is not an exact tumoral marker [6]. Many men may have PCa in spite of low serum levels of PSA [7]. PSA concentration is higher than 4 ng/mL in 25% to 50% of patients with benign prostatic hyperplasia. Lack of specificity is the main drawback of PSA, and therefore other approaches to the problem have been developed [8]. For every one in five biopsies avoided, 5% of existing tumours go undiagnosed. With PSA from 4 to 10 ng/mL, 75% of biopsies are negative (90% sensitivity). Free PSA adds 15% to 25% specificity. In the year 2000, of 15 million PSA tests in the United States, 85% were normal and 15% were abnormal. One million biopsies were carried out, of which a third were positive (333.333 patients), according to data from Parkin [9].

The proportion of free PSA to total PSA, i.e., the free PSA/total PSA index or ratio (iPSA), indicates the quantity of free-circulating PSA in comparison to PSA bound to proteins. Several studies assert that iPSA is lower in patients with PCa than in those with benign prostatic hyperplasia (BPH). Age-adjusted probability of PCa is more closely correlated with iPSA than with PSA alone, and is also more effective in differentiating between PCa and benign conditions [10]. iPSA may be affected by the instability of free PSA at 4 °C and ambient temperature, by variable study conditions, or by concomitant BPH [11]. When prostate biopsies are done in men with PSA between 4–10 ng/mL, in the case of having iPSA < 0.10, the probability of being positive for cancer is 56%, and when iPSA > 0.25, they are positive for cancer alone in 8% of cases [12]. Unnecessary biopsies can be reduced by 20% when iPSA (with a cut-off of 25%) is taken into consideration. iPSA may, therefore, prove useful in PCa diagnosis and screening in patients with a PSA of 4–10 ng/mL [10].

There is research on the usefulness of PSA proenzymes (proPSA). In men with a PSA of 2–10 ng/mL, the ratio of proPSA to free PSA was shown to be more specific for detecting PCa than total PSA or free PSA alone [13]. Other authors have also noted that with PSA 2.5–5 ng/mL, the ratio of proPSA to free PSA was more effective at detecting PCa and preventing unnecessary biopsies [14].

There are two principal measures of PSA kinetics: PSAV (absolute yearly increase in serum PSA (ng/mL/year) [15]) and PSA duplication time (PSA-DT: measures the exponential increase in serum PSA over time [16]). PSA velocity (PSAV), the variation in total serum PSA over time, may be helpful for diagnosing PCa, with adjustments according to prostate volume, presence of BPH, variability of the interval between measurements, and PSA acceleration or deceleration over time [17]. PSAV was first described by Carter in 1992 [18]. It is usually defined as the absolute yearly annual increase in PSA, measured in ng/mL/year [16]. When monitoring patients in whom PCa is suspected and with a cut-off point of ≥0.3–0.5 ng/mL/year, the specificity of the test became 90% with PSAV, compared with 60% if only total PSA were used. However, the sensitivity of PSAV was not an improvement over total PSA, at least in early studies, with few patients [19].

The aim of this study was to establish the association of changes of PSA velocity, free PSA/total PSA index or ratio, and prostate condition (healthy, benign prostatic hyperplasia, prostatitis, prostatic intraepithelial neoplasia, prostate cancer) in a priori healthy males.

## 2. Methods

### 2.1. Design

This was a prospective multicentre observational study of 2035 men over 47, a priori in good health, who were offered a prostate health check through social media. The Health Area from which individuals from the community were recruited was the province of Salamanca (Spain).

The offer or communication of the existence of the study was made to the general public by social media, such as the general newspaper, the only newspaper in the health area. All men in the community were informed equally. Patients were excluded if they were already diagnosed with a prostate pathology. Therefore, the population of origin of the patients is the same: it is a standard Spanish health area, with 340,000 inhabitants. These individuals had not previously consulted in Primary Care Health System for prostatic symptoms or other types of urinary symptoms.

A routine protocol was followed for diagnosing prostate health: anamnesis, physical examination, and relevant complementary exams.

The monitoring of the individuals has been from 1 February 2019 to 1 September 2020.

The individuals are men from the community who want to know the state of their prostate. At the consultation, all individuals were questioned about prostate symptoms. Anamnesis, physical examination and determination of PSA, free PSA, and PSA velocity were performed in all individuals. All males had more than one PSA determination that allowed to know the velocity of PSA and direction of the acceleration. According to the results of the anamnesis, physical examination and PSA, further studies were requested, such as a prostate biopsy, or the patient was informed that he was healthy and did not require further studies.

All individuals were recruited to participate in the study between 1 February 2019 and 30 March 2019. A rigorous health study was conducted in all of them to reach a definitive diagnosis of how their prostate was.

All patients were reassessed at various medical visits:

In the first visit, anamnesis, physical examination and determination of PSA, free PSA, and PSA velocity were performed.

After 7 days, they were re-evaluated to know the results of the tests.

In those with suspicion of malignancy, the prostate biopsy was performed within a month. Afterward, treatment was established according to the outcome of cancer or no cancer.

All patients were subsequently reassessed every 3 months until 1 September 2020.

No new cancers have appeared in follow-up.

A total of 435 ultrasound-guided transrectal prostate biopsies were performed, with 6 cylinders per prostate lobe.

Biopsy was indicated in the following cases:(1)PSA greater than 4 ng/mL and free PSAl/total PSA index less than 15%;(2)Suspicious rectal examination;(3)Positive PSA velocity higher than 0.75 ng/mL in 12 months.

The entire study was coordinated by Dr. María Fernanda Lorenzo-Gómez, who is a urologist. Dr. Lorenzo-Gómez explained the need for a biopsy to all the patients on whom it was performed. The entire process of urological management of cancer results was carried out in the urology consultation of Dr. Maria Fernanda Lorenzo Gomez at the University Hospital of Salamanca (37007 Salamanca, Spain). However, when the results were not cancer, the report of the biopsy results was made by the Primary Care Physicians participating in the research, in the Primary Care Center that corresponded to the individuals in their respective Health Areas.

Prostate MRI was not performed in this study.

In 50 patients, repeated PSA determinations were performed to corroborate PSA, with a difference of 7 days between the determination taken into account for the study and the repeated determination for corroboration. There was no difference between the reference determination and the corroboration determination. No laboratory errors were found.

Prostate biopsies were performed by the routine urologists corresponding to each patient. That is, once the need for a biopsy was decided to determine the existence of prostate cancer, the routine urologist for each patient was the one who performed the biopsy.

Prostate biopsies were performed in 260 individuals of GA, 111 individuals of GB, and 64 individuals of GC.

### 2.2. Groups by Serum PSA Level

To decide the study groups, two factors were considered:-The value of PSA in all of the 2035 individuals: mean 2.01, SD 10.01, median 1.1, range 0.04–435.94;-The distribution of the values: on one hand, it was tried that the groups had a number of individuals as similar as possible. On the other hand, that the value of the PSA figures could have a clinical significance.

GA (*n* = 518): men with serum PSA above or equal to 2.01 ng/mL.

GB (*n* = 775): men with serum PSA above or equal to 0.78 ng/mL and below 2.01 ng/mL.

GC (*n* = 742): men with serum PSA below 0.78 ng/mL.

Inclusion criteria: men who wish to be informed on the health of their prostate.

Exclusion criteria: men with a previously diagnosed prostate condition.

### 2.3. Variables

The variables were prostate-specific antigen (PSA); age (years); body mass index (BMI); PSA velocity (PSAV) (ng/mL per year); free PSA/total PSA index (iPSA); direction of PSAV acceleration (increasing: positive, or decreasing: negative); prostate diagnosis (benign prostatic hyperplasia (BPH), prostatic intraepithelial neoplasia (PIN), or infectious and non-infectious prostatitis and prostatic adenocarcinoma (PCa)); de novo diagnoses of urinary tract abnormalities; and concomitant treatments, diseases and conditions. The diagnosis of the prostate situation was made according to the ordinary practice of both primary care physicians and urology specialists: anamnesis, physical examination, routine complementary examinations of the study of prostate health. The PSA velocity is the value of the PSA change over time, adjusted to 12 months. It was called positive acceleration when the velocity was positive, that is, when the PSA value was increasing. It was called negative acceleration when the speed was negative, that is, when the PSA value was decreasing.

### 2.4. Statistical Analysis

The automatic statistics calculator NSSS2006/GESS2007 was used. Results were analysed with descriptive statistics, Student’s t, Chi2, Fisher’s exact test, ANOVA (with Scheffe’s test for normal samples and Kruskal–Wallis for other distributions), and multivariate analysis. For the multivariate analysis, two highly rigorous statistical techniques were used for this type of distribution, that is, to analyse individuals from the community with great differences, despite being all of the same race, culture, lifestyle and same Area of Health. First, a multidimensional scaling analysis was carried out with the non-metric PROXSCAL method, which allows spatial representation, in the form of a map, and shows the proximity between the variables and the analysis of the relationship between them. A stress analysis of the model itself was carried out to check its usefulness in this analysis and in this population, resulting in that the applied model was perfect and adjusted well to the variables in each dimension, with a normalized gross stress of 0.5%. The model showed that the number of dimensions to represent the variables was three, being the most appropriate setting for the representation. The model displays the variables and their distances (coordinates) within each dimension, which determines the representation of each variable in the dimensions. The other multivariate analysis method used was the vector model. The objective of this model is to represent the variables in the investigated spaces or dimensions: it shows the projection of the variables and the correlation between the values of each one and the projection in the dimensions between the groups. This multivariate analysis includes an adjustment of the variables for age. Statistical significance was accepted for *p* < 0.05.

### 2.5. Ethical Considerations

All patients gave signed consent for inclusion in the study. The study was conducted in accordance with Directive 2001/20/EC of the European Parliament and of the Council of 4 April 2001, and in compliance with the standards of good clinical practice of the Ministry of Health and Consumer Affairs and the Spanish Agency of Medicines and Medical Devices [20].

The study protocol 2019/0102-MLG-GRUMUR was approved by the Ethics Committee for Clinical Research of the University Hospital of Ávila, Spain.

## 3. Results

Clinical and pathological variables of the whole sample are summarised in Table 1. Mean age was 62.35 years (SD 8.12), median 61, range 47–94; age was lowest in GC (*p* = 0.0001). Mean BMI was 27.89 kg/m^2^ (SD 3.96), median 27.58, range 18.56–57.13; there were no differences between groups (*p* = 0.07556). Mean PSAV was 0.69, SD 2.16, median 0.13, range 0.001–34.46; PSAV was lowest in GC (*p* = 0.0001) (Figure 1). Mean PSA index was 27.39 u/L (SD 14.25), median 24.29, range 3.7–115; PSA index was lowest in GA (*p* = 0.0001). PSAV had more positive acceleration in GA and more negative acceleration in GC (*p* = 0.00001). No patient for whom a prostate biopsy was indicated refused to perform it. The 435 biopsies were done following the same protocol of antibiotic protection and intestinal preparation, in addition to applying rectal povidone-iodine before and after the biopsy. No patient presented complications that required hospital admission. There were no patients with significant bleeding or excruciating pain. Twenty patients presented with low-grade fever in the 24 h after the biopsy, which required extending the antibiotic coverage to aminoglycosides (tobramycin 100 mg intramuscular injection every 24 h for 6 days) and prolonging the oral antibiotic (usually amoxicillin 875 mg every 8 h orally, or fosfomycin 500 mg every 8 h orally) for 10 days.

There were 1600 (78.62%) cases of normal prostate or BPH, 322 (15.82%) cases of PIN or non-infectious prostatitis, and 113 (5.55%) cases of PCa. All patients in whom PIN, prostate cancer or non-infectious prostatitis were diagnosed, the diagnosis was made by prostate biopsy. In patients in whom a biopsy was not indicated, all patients met the clinical criteria for benign prostatic hyperplasia defined by the American Urological Association in 1992, a clinical definition that is used today [21].

There were more cases of BPH in GC and more cases of PIN or prostatitis and cancer in GA (*p* = 0.00001). Mean Gleason was 6.02 (SD 1.25), median 6.00, range 4–8.

In the cohort of prostate cancer diagnosis (Table 2), mean age was 67.30 years (SD 8.11), median 66.00, range 53–86; there were no differences (*p* = 1.000). Mean BMI was 26.97 kg/m^2^ (SD 3.35), median 27.00, range 20.00–36.00; there were no differences between groups (*p* = 0.762). Mean PSAV was 3.13, SD 5.41, median 1.33, range 0.001–34.46; there were no differences between groups (*p* = 0.828). Mean PSA index was 17.16 u/L (SD 8.85), median 15.03, range 5.03–48.53; PSA index was lowest in GA (*p* = 0.0041). PSAV had more positive acceleration in GA and more negative acceleration in GC (*p* = 0.0036). Mean Gleason was 7.30 (SD 0.755), median 7.00, range 5–8, was lowest in GC (*p* = 0.00004).

De novo diagnoses of the urinary tract: 5 cases of urinary incontinence (UI), 1 case of stress UI, 9 mixed UI, 16 discomfort/pain in the lower urinary tract, 112 cases of voiding disorders (unclassifiable, not requiring medical treatment), 12 urethral strictures, 19 haematuria, 51 cystitis, 3 pyelonephritis, 4 pelvic inflammatory disease. No differences were found between groups (*p* = 0.364).

Concomitant conditions and diseases were also investigated. A total of 801 patients (39.36%) with high blood pressure (HBP), 304 (14.93%) with diabetes mellitus (DM), 905 (44.47%) with dyslipidaemia, 242 (11.89%) with other metabolic disorders, 208 (10.22%) with anxiety, 119 (5.84%) with depression, 385 (18.91%) with various pain complaints, 298 (14.64%) with ear, nose and throat disorders (ENT), 250 (12.28%) with allergies, and 879 (43.19%) with self-limiting conditions were identified. There were more cases of hypertension (46.52%), dyslipidaemia (46.71%) and other metabolic disorders (12.54%), and ENT disorders 18.53%) in GA than in GC, where there were more cases of DM (15.76%), anxiety (10.51%), depression (6.33%), various pain complaints (19.94%), other self-limiting conditions (43.93%), and allergies (12.93%); in GB, no concomitant condition or disorder predominated (*p* = 0.0212). The following cases of toxic substance use were reported: 603 (29.63%) no substance use, 24 (1.17%) non-smokers, 300 (14.74%) active smokers, 7 (0.34%) ex-smokers, and 381 (18.72%) drinkers, with no differences between groups (*p* = 0.512).

There was a difference in concomitant treatments (*n*/%): GA had more use of step-two analgesics (21/4.05), ARA2 antihypertensives (92/17.76), diuretics (56/10.81), alpha-blocker antihypertensives (74/14.28), and omeprazole (84/16.21); GB had more use of metformin (85/10.96), ACEI antihypertensives (106/13.67), multiple pharmaceutical products (190/24.51), and acetylsalicylic acid (27/3.42); and GC had more benzodiazepine (55/7.41), antipsychotics (25/3.36), absence of concomitant treatments (55/7.41), sitagliptin (29/3.90), insulin (25/3.36), step-one analgesics (96/12.93), lipid-lowering agents (231/31.13), and gastro-protectant H2-receptor antagonists (such as ranitidine or famotidine) (36/4.85) (*p* = 0.0004).

The multivariate analysis (non-metric proximity scaling, PROXSCAL) shows that the variables which correlate most with the condition of the prostate are PSAV and the direction of PSAV acceleration (positive or negative) (*p* = 0.0051). The PSA index is correlated to the presence of prostatitis, PCa, and BPH (Figure 2).

In the multivariate analysis with the vector model, the variables with the highest weight and correlation to prostate health in GA are prostatitis, PCa, treatment with alpha-blockers and a history of previous smoking (*p* = 0.0097). In GB, prostate health is correlated with BPH, concomitant treatments and their respective conditions, and positive acceleration of PSAV (*p* = 0.0071). In GC, prostate health is correlated with BPH and use of ranitidine (*p* = 0.047) (Figure 3).

## 4. Discussion

Mean PSA velocity in this study was 0.69 ng/mL/year, with a median of 0.13 and a range of 0.001–34.46. This is a high PSAV, given that an increase of over 0.75 ng/mL/year over the baseline suggests PCa [7]. Direction of PSAV was positive or increasing in 54.92% of men, compared to 45.08% where velocity was descending. It is of great clinical relevance, in a multivariate study where age was not a bias, that PSAV was lowest in GC and highest in GA. Additionally, there was more positive PSAV (69.16%) in GA and GB (69.16% and 57.54% respectively) and more negative PSAV in CG (58.72%). Although other studies have questioned the usefulness of PSAV when compared with PSA value alone [22], our study shows the prognostic information given by both PSA velocity and the direction of its acceleration at PSA thresholds as low as 0.78 and 0.21 ng/mL, which is totally novel.

Average PSAV is high compared to other articles. It has been shown that in comparison with other Spanish, European and world registries, the adjusted incidence rates of prostate cancer in the province of Salamanca (37007 Spain) are lower than those registered in the northern and western provinces of Spain (except Vizcaya), northern and western European countries, USA and Canada, and higher than those registered by the southern provinces of Spain, some southern and eastern European countries, and part of central Europe [23]. Therefore, these finding are the subject of future research to clarify the significance of this high PSAV.

Although other studies have focused on the relationship between PSA velocity and prostate cancer, our study provides important novelties. The first is that it has been carried out with a completely new design: the study has been carried out on men from the community who, when faced with a public announcement in a social communication medium, such as the newspaper in the written press, which covers all the population and has a homogeneous and complete diffusion throughout the geography of the Salamanca Health Area (La Gaceta de Salamanca). This is a fact that makes the inclusion of participating individuals original. Therefore, this investigation of PSA velocity and acceleration is very interesting. The second novelty of impact is that the direction of the acceleration of the PSA is considered: positive or negative, in the population group of the community. And the third novelty is that it is not a study on the relationship between PSAV and prostate cancer, but rather a study between PSAV and any prostate condition: BPH, PIN, non-infectious prostatitis or cancer. This is completely new.

Until now, PSAV had only been investigated to determine whether it was related to the risk of prostate cancer diagnosis, the indication for treatment in men under active surveillance, or the ability to predict recurrence of prostate cancer [24]. In our study, this is the first time that PSAV has been used to investigate any type of prostate pathology, including benign ones.

Indeed, our study reports not only on the relationship of PSAV with prostate cancer that other authors have investigated [22]. Our research shows a relationship between changes in PSAV and both benign and malignant pathology, therefore, providing new scientific information.

Although the objectives of the study are to know the relationship of changes in PSAV and iPSA with the main diagnosis of prostate status, it is convenient to take into account circumstances such as medication or concomitant diseases that could influence the results. Therefore, all diagnoses have been investigated, both urinary system and other parts of the body and the concomitant medications.

Ranitidine, like famotidine, belongs to the group of histamine antagonist drugs at type II histamine receptors, also called H2. They are used in the treatment of gastrointestinal diseases such as gastric and duodenal ulcer, gastroesophageal reflux, and other pathological hypersecretory conditions [25]. One of the effects of H2 antagonists is hyperprolactinemia [26]. The relationship between testosterone level and prostate disease is controversial. While some authors claim that there is no association between prostate cancer and serum levels of testosterone and prostate antigen [27], other authors claim that testosterone is significantly lower in patients with prostate cancer than in controls [28]. Therefore, it is an active research area to find the relationship between testosterone levels and prostate disease.

### PSA Index (iPSA)

Mean iPSA was 27.39%, with a median of 24.29 and a range of 3.7–115; iPSA was lowest in GA (mean 20.65%, SD 9.46, median 18.68, range 3.7–75.85) and highest in GC (mean 33.57%, SD 16.51, median 29.63, range 6.25–115).

A PSA index below 15% is currently thought to be associated with cancer; an index above 20–25% is associated with benign conditions [1]. Our results are coherent with this.

However, we highlight that in the multivariate analysis, iPSA trended downward in the group with more PCa, and upward where PSA levels were lowest, in ranges as low and narrow as 0.78 ng/mL and 2.01 ng/mL. On the other hand, we have found that an independent factor, the use of gastro-protectant H2-receptor antagonists (such as ranitidine or famotidine) did not correlate with PCa but was correlated with low PSA and low iPSA.

The individuals are very different, and a powerful statistical analysis is necessary. For this reason, an analysis specially designed for this type of population has been carried out, such as the non-metric proximity scaling PROXSCAL. Because it is a high precision statistical analysis for samples with very different individuals, such as the population of this study, it is the main analysis. On the other hand, we have added a conventional ANOVA analysis.

Our study in male individuals from the community demonstrates the importance of changes in PSA velocity and the influence of treatments that modify testosterone levels in a chronic and sustained way, such as ranitidine-type gastric protectors. These findings make us progress in these lines of research in the future.

We believe that our study can modify routine determination habits of iPSA and PSAV. Thus far, the iPSA has not been determined in our environment if the PSA is less than 4 ng/mL. Our study shows that it is convenient to determine the iPSA at PSA values lower than 4 ng/mL since iPSA provides relevant clinical information on PSA lower than 4 ng/mL. Regarding the speed of PSA, it also advises changes in the usual clinical practice of the management of prostate disease, since in our study we observed that changes in PSAV, increasing or decreasing, provide information on whether the prostate disease is benign or malignant.

## 5. Conclusions

In a priori healthy men over 47 who are interested in prostate health, patients with high PSA are more likely to undergo biopsy. In this group, PSA velocity greater than 0.75 ng/mL is associated with prostate cancer diagnosis. PSA velocity, the direction of PSAV acceleration and the free PSA/total PSA index, at cut-off points of PSA 0.78 and 2.01 ng/mL, predict the probability of benign or malign pathology of the prostate.

## Figures and Tables

**Figure 1 jcm-09-03400-f001:**
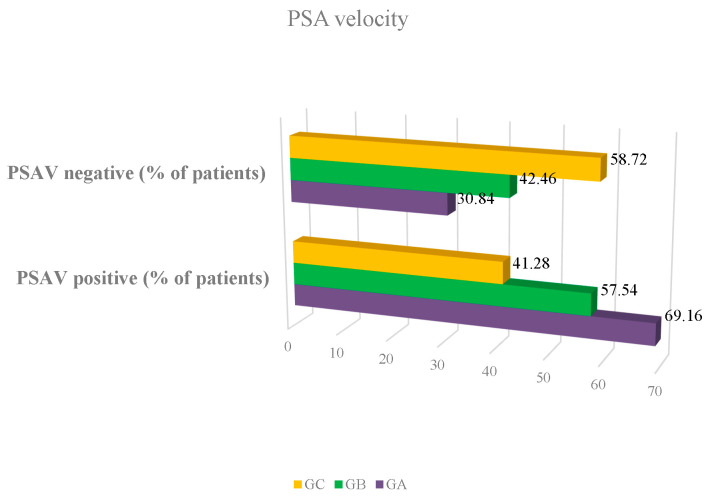
Percentage of patients with PSA velocity negative or positive. PSA: prostate-specific antigen; GA: men with PSA ≥ 2.01 ng/mL; GB: men with PSA ≥ 0.78 and <2.01 ng/mL; GC: men with PSA < 0.78 ng/mL. PSA velocity negative: the value of the PSA decreases over time; PSA velocity positive: the value of PSA increases over time.

**Figure 2 jcm-09-03400-f002:**
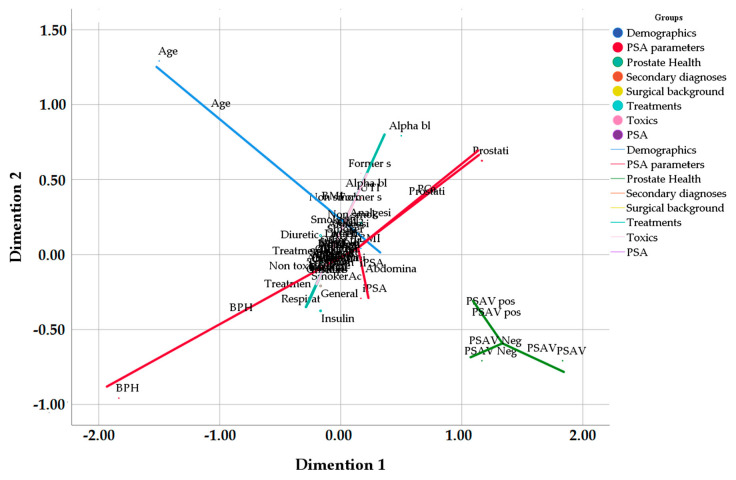
Relationship between variables and prostate condition. Dimension 1: dimension 1 of the PROXSCAL multivariate analysis. Dimension 2: dimension 1 of the PROXSCAL multivariate analysis. BMI: body mass index; iPSA: prostate-specific antigen index; BPH: benign prostatic hyperplasia; PCa: prostate cancer; PSAV: velocity of PSA; UTI: urinary tract infections; ORL: otorhinolaryngological disorder; AHT: arterial hypertension; DM2: type 2 diabetes mellitus; ACEI: angiotensin conversion enzyme inhibitor: LMWI: low molecular weight insulin; BZD: benzodiazepine; ASA: acetylsalicylic acid.

**Figure 3 jcm-09-03400-f003:**
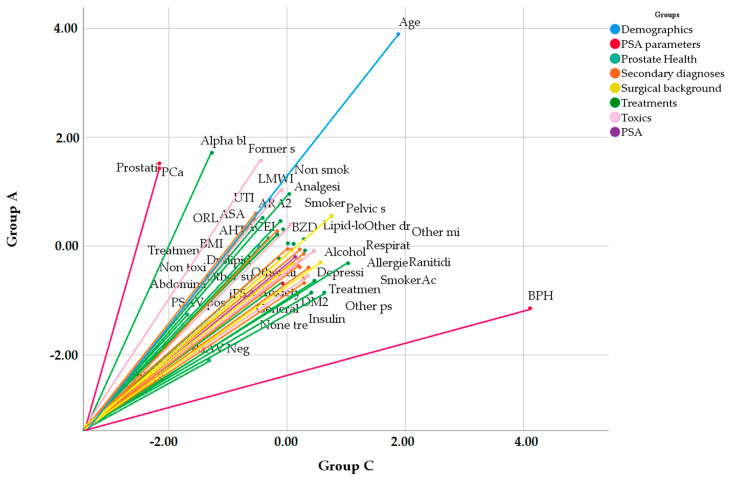
Vectorial analysis of relationship between variables and prostate condition in GA y GC. GA: men with PSA ≥ 2.01 ng/mL; GC: men with PSA < 0.78 ng/mL; PSA: prostate-specific antigen; BMI: body mass index; iPSA: prostate-specific antigen index; BPH: benign prostatic hyperplasia; PCa: prostate cancer; PSAV: velocity of PSA; UTI: urinary tract infections; ORL: otorhinolaryngological disorder; AHT: arterial hypertension; DM2: type 2 diabetes mellitus; ACEI: angiotensin conversion enzyme inhibitor: LMWI: low molecular weight insulin; BZD: benzodiazepine; ASA: acetylsalicylic acid.

**Table 1 jcm-09-03400-t001:** Variables and diagnosis in men who were given prostate exams.

Variable	Group						*p*
	GA		GB		GC		
	Mean (SD)	Median (Range)	Mean (SD)	Median (Range)	Mean (SD)	Median (Range)	
Age	65.80 (8.24)	65 (50–94)	62.24 (7.68)	62 (47–93)	60.07 (7.66)	58 (49–94)	0.0001
BMI	27.49 (3.61)	27.18 (19.75–46.25)	27.93 (4.19)	27.68 (18.56–57.13)	28.06 (3.90)	27.68 (19.41–43.52)	0.0755
PSAV	1.56 (3.53)	0.36 (0.0013–34.46)	0.37 (1.02)	0.11 (0.001–17.88)	0.36 (1.36)	0.057 (0.001–20.96)	0.0001
iPSA	20.65 (9.46)	18.68 (3.7–75.85)	26 (11.93)	23.55( 6.02–75.78)	33.57 (16.51)	29.63 (6.25–115)	0.0001
PSAV acceleration	Positive (*n*/%)	Negative (*n*/%)	Positive (*n*/%)	Negative (*n*/%)	Positive (*n*/%)	Negative (*n*/%)	0.00001
	305/69.16	136/30.84	355/57.54	262/42.46	239/41.28	340/58.72	
PRIMARY DIAGNOSIS	*n*	%	*n*	%	*n*	%	*p*
PD: BPH	258	48.81	664	86.68	678	91.32	0.00001
PD: PIN—non-infectious prostatitis	158	30.50	106	13.68	58	7.82	
PD: PCa	102	19.69	5	0.65	6	0.81	
Gleason	7.28 (0.80)	7.00 (6–8)	6.53 (0.50)	7.00 (6–7)	4.62 (0.48)	5.00 (4–5)	0.006

PSA: Prostate-specific antigen; GA: men with PSA ≥ 2.01 ng/mL; GB: men with PSA ≥ 0.78 and <2.01 ng/mL; CG: men with PSA < 0.78 ng/mL; BMI: body mass index; PSAV: speed of PSA; iPSA: free PSA/total PSA index or ratio; PD: principal diagnosis; BPH: benign prostatic hyperplasia; PIN: prostatic intraepithelial neoplasia; PCa: prostate cancer.

**Table 2 jcm-09-03400-t002:** Main variables in men with prostate cancer diagnose in the groups.

Variable	Group						*p*
	GAP Ca+		GB PCa+		GC PCa+		
	Mean (SD)	Median (Range)	Mean (SD)	Median (Range)	Mean (SD)	Median (Range)	
Age	67.59 (8.16)	66.50 (53.00–86.00)	62.40 (7.76)	65.00 (55.00–72.00)	66.33 (7.20)	68.00 (53.00–74.00)	1.0000
BMI	26.85 (3.38)	26.00 (20.00–36.00)	29.30 (2.00)	29.00 (27.00–31.00)	27.33 (3.50)	26.50 (23.00–32.00)	0.762
PSAV	3.21 (5.49)	1.38 (0.001–34.46)	0.20 (0.17)	0.19 (0.07–0.32))	2.25 (0.20)	2.20 (2.20–2.35)	0.828
iPSA	16.40 (8.39)	14.76 (5.03–48.53)	21.55 (13.87)	16.35 (7.06–39.29)	26.41 (6.67)	26.39 (18.20–34.94)	0.0041
PSAV acceleration	Positive (*n*/%)	Negative (*n*/%)	Positive (*n*/%)	Negative (*n*/%)	Positive (*n*/%)	Negative (*n*/%)	0.0036
	76/74.5	26/25.50	2/30.00	3/70.00	1/16.70	5/83.30	
Gleason	7.47 (0.50)	7.00 (7–8)	6.60 (0.54)	7.00 (6–7)	5.10 (0.10)	5.00 (5–5.10)	0.00004

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
