# Peer review of "The Association between Prostate-Specific Antigen Velocity (PSAV), Value and Acceleration, and of the Free PSA/Total PSA Index or Ratio, with Prostate Conditions"

_jcm, 2020, doi:10.3390/jcm9113400_

Round 1
Reviewer 1 Report
Dear authors,
Thank you kindly for the opportunity to review this manuscript. Congratulations on completing a prospective study involving a large number of patients. There are some significant improvements that can be made to this paper for publication.
First is to acknowledge the deficiencies in comparing three groups with very different types of men. Due to the low levels of prostate cancer diagnosis and prostate biopsy in GB and GC, it cannot be compared by conventional ANOVA. I would recommend statistical review to allow for appropriate comparison and reporting.
Secondly, more information is required in general to allow for appropriate reporting of this trial. How many patients declined prostate biopsy? What were the complications? Who and how were the biopsies reported? Was mpMRI involved? Were there any extra PSA readings outside of the study? Who performed the biopsies and how many from the each group underwent biopsies? What were the gleason grades and positivities?
I have made a few more points below.
Page 1Line 15 Abstract: Introduction - normal cut-off point is more conventionally 0.3-0.5ng/ml
Page 1 Line 17 In complete sentences throughout the abstract: The aim of the study is to establish…
Page 2 Line 66 third of a million is not 250.000 patients.
Page 2 Design - the study should emphasise the homogenous and specific source of the patients. These patients who are healthy were concerned enough to have physical examination, PSA tests and biopsies. The study may not be generalisable.
Table 1. Keep decimals consistent - full stop vs comma
Table 1. Only 5 and 6 patients were diagnosed with GB and GC. Statistical analysis between these groups are not appropriate. It is also important to acknowledge the number of biopsies performed per group.
Page 3 Line 161. Without biopsy in everyone, it is not reasonable to state that patients had normal prostate, BPH or non-infectious prostatitis. It can only be stated that 113 patients were diagnosed with PCa based on 435 biopsies.
Page 3 Line 125. Biopsy was performed in patients with PSA greater than 4. Naturally, the majority of the patients had biopsy in the GA with the highest PSA values.
Page 4 Line 172. De novo diagnoses of urinary tract abnormalities.
Results. Average PSAV is high compared to other articles. Are there reasons for this?
Figure 1. axis units are missing. What are the numbers and wordings eluding to?
Page 8. Conclusion.
I do not agree with the conclusion. From the information provided, this study demonstrates that in patients who are interested in prostate health, patients with high PSA are more likely to undergo biopsy. In this group, PSA velocity greater than 0.75ng/ml was associated with prostate cancer diagnosis, although the baseline data is required to conclude this.
There would be extensive data from this cohort, and I believe with some modifications, this would be suitable for publication. We look forward to seeing the next version.
Kind regards.
Author Response
ANSWERS TO REVIEWER 1 WE PUT THEM IN GREEN COLOR:
Dear authors,
Thank you kindly for the opportunity to review this manuscript. Congratulations on completing a prospective study involving a large number of patients. There are some significant improvements that can be made to this paper for publication.
1.-First is to acknowledge the deficiencies in comparing three groups with very different types of men. Due to the low levels of prostate cancer diagnosis and prostate biopsy in GB and GC, it cannot be compared by conventional ANOVA. I would recommend statistical review to allow for appropriate comparison and reporting.
ANSWER:
The reviewer is correct: that individuals are very different and a powerful statistical analysis is necessary. For this reason, an analysis specially designed for this type of population has been carried out, such as thenon-metric proximity scaling PROXSCAL. Because it is a high precision statistical analysis for samples with very different individuals, such as the population of this study, it is the main analysis. But on the other hand, in the medical sciences it is a little known analysis, for which the authors did not want to risk that the results were not well understood with the PROXSCAL analysis and therefore we have added a conventional ANOVA analysis, which we agree with the reviewer, as the only analysis, it would not be enough.
We add this paragraph in discussion:
The individuals are very different and a powerful statistical analysis is necessary. For this reason, an analysis specially designed for this type of population has been carried out, such as thenon-metric proximity scaling PROXSCAL. Because it is a high precision statistical analysis for samples with very different individuals, such as the population of this study, it is the main analysis. On the other hand, we have added a conventional ANOVA analysis.
2.-Secondly, more information is required in general to allow for appropriate reporting of this trial. How many patients declined prostate biopsy? What were the complications? Who and how were the biopsies reported? Was mpMRI involved? Were there any extra PSA readings outside of the study? Who performed the biopsies and how many from the each group underwent biopsies? What were the gleason grades and positivities?
ANSWER:
We add this paragraph in the results:
No patient for whom a prostate biopsy was indicated refused to allow it to be performed.
3.-What were the complications?
We add this paragraph in the results:
The 435 biopsies that were performed were done under the same protocol of antibiotic protection and intestinal preparation, in addition to applying rectal povidone-iodine before and after the biopsy. No patient presented complications that required hospital admission. There were no patients with significant bleeding or excruciating pain. In 20 patients there was a low-grade fever in the 24 hours after the biopsy, which required extending the antibiotic coverage to aminoglycosides (tobramycin 100 mg intramuscular injection every 24 hours for 6 days) and prolonging the oral antibiotic (usually amoxicillin 875 mg every 8 hours orally, or fosfomycin 500 mg every 8 hours orally) for 10 days.
4.-Who and how were the biopsies reported?
We add this paragraph in Methods:
The entire study was coordinated by Dr. María Fernanda Lorenzo Gomez, who is urologist. Dr. Lorenzo was who explained to all the patients in whom it was performed, the need for a biopsy. The entire process of urological management of cancer results was carried out in the Urology consultation of Dr. Maria Fernanda Lorenzo Gomez at the University Hospital of Salamanca (37.007 Salamanca, Spain). However, when the results were not cancer, the report of the biopsy results was made by the Primary Care Physicians participating in the research, in the Primary Care Center that corresponded to the individuals in their respective Health Áreas.
5.-Was mpMRI involved?
We add this paragraph in Methods:
Prostate MRI was not performed in this study.
6.-Were there any extra PSA readings outside of the study?
We add this paragraph in Methods:
In 50 patients, repeated PSA determinations were performed to corroborate PSA, with a difference of 7 days between the determination taken into account for the study and the repeated determination for corroboration. There was no difference between the reference determination and the corroboration determination. No laboratory errors were found.
7.- Who performed the biopsies and how many from the each group underwent biopsies?
We add this paragraph in Methods:
Prostate biopsies were performed by the routine urologists corresponding to each patient. That is, once the need for a biopsy was decided to determine the existence of prostate cancer, the routine urologist for each patient was the one who performed the biopsy.
Prostate biopsies were performed in 260 individuals of GA, 111 individuals of GB, and 64 individuals of GC.
8.-What were the gleason grades and positivities?
In GA there were 102 cancer and 158 PIN or non-infectious prostatitis, in GB there were 5 cancer and 106 PIN or non-infectious prostatitis and in CG there were 6 cancer and 58 PIN or non-infectious prostatitis, as shown in Table 1.
We add Gleason results in the text and in the table 1.
9.-I have made a few more points below.
Page 1Line 15 Abstract: Introduction - normal cut-off point is more conventionally 0.3-0.5ng/ml
ANSWER:
WE CHANGE WHERE IT SAYS: 0.75-1 ng/ml
BY: 0.3-0.5 ng/ml
10.-Page 1 Line 17 In complete sentences throughout the abstract: The aim of the study is to establish…
ANSWER:
WE CHANGE WHERE IT SAYS: To establish
BY: The aim of the study is to establish
11.-Page 2 Line 66 third of a million is not 250.000 patients.
ANSWER:
WE CHANGE WHERE IT SAYS: 250.000
BY: 333.333
12.-Page 2 Design - the study should emphasise the homogenous and specific source of the patients. These patients who are healthy were concerned enough to have physical examination, PSA tests and biopsies. The study may not be generalisable.
ANSWER:
Study design: the offer or communication of the existence of the study was made to the general public by social media, such as the general newspaper, the only newspaper in the health area. All men in the community were informed equally. The exclusion factor was that they were already diagnosed with a prostate pathology. Therefore, the population of origin of the patients is the same: it is a standard Spanish health area, with 340,000 inhabitants. These individuals had not previously consulted in Primary Care Health System for prostatic symptoms or other types of urinary symptoms.
13.-Table 1. Keep decimals consistent - full stop vs comma
ANSWER:
We put all the decimals in the same way: full stop.
14.-Table 1. Only 5 and 6 patients were diagnosed with GB and GC. Statistical analysis between these groups are not appropriate. It is also important to acknowledge the number of biopsies performed per group.
ANSWER:
This aspect is already answered above: an appropriate analysis was chosen for the multivariate analysis, but we wanted to add a conventional analysis so as not to risk that the results would not be clear. And also the number of biopsies in each group has already been answered above.
15.-Page 3 Line 161. Without biopsy in everyone, it is not reasonable to state that patients had normal prostate, BPH or non-infectious prostatitis. It can only be stated that 113 patients were diagnosed with PCa based on 435 biopsies.
ANSWER:
We add this paragraph:
All patients in whom PIN, prostate cancer or non-infectious prostatitis were diagnosed, the diagnosis was made by prostate biopsy. In patients in whom a biopsy was not indicated, all patients met the clinical criteria for benign prostatic hyperplasia defined by the American Urological Association in 1992, a clinical definition that is used today (AÑADIR REFERENCIA: Barry M.J., Fowler F.J., Jr, O’Leary M.P., Bruskewitz R.C., Holtgrewe H.L., Mebust W.K. The American Urological Association symptom index for benign prostatic hyperplasia. The Measurement Committee of the American Urological Association. J Urol. 1992;148:1549–1557. [PubMed] [Google Scholar].
16.-Page 3 Line 125. Biopsy was performed in patients with PSA greater than 4. Naturally, the majority of the patients had biopsy in the GA with the highest PSA values.
ANSWER:
The reviewer is correct that one of the assumptions to indicate biopsy was the PSA value, but the digital rectal examination findings and PSA velocity were also taken into account. It cannot be ignored that a fundamental factor in the clinical management of prostate pathology is the total PSA value.
17.-Page 4 Line 172. De novo diagnoses of urinary tract abnormalities.
ANSWER:
WE CHANGE WHERE IT SAYS: de novo diagnoses of urinary tract diseases or conditions
BY: de novo diagnoses of urinary tract abnormalities.
18.-Results. Average PSAV is high compared to other articles. Are there reasons for this?
ANSWER:
We add this paragraph in Discussion:
Average PSAV is high compared to other articles. It has been shown that in comparison with other Spanish, European and world registries, the adjusted incidence rates of prostate cancer in the province of Salamanca (37007 Spain) are lower than those registered in the northern and western provinces of Spain (except Vizcaya), northern and western European countries, USA and Canada, and higher than those registered by the southern provinces of Spain, some southern and eastern European countries, and part of central Europe (Javier San-Bartolomé-Gutiérrez2017). Therefore, these finding are the subject of future research to clarify the significance of this high PSAV.
19.-Figure 1. axis units are missing. What are the numbers and wordings eluding to?
ANSWER: We change the text of the figure by:
Percentage of patients with PSA velocity negative or positive. PSA: prostate-specific antigen; GA: men with PSA ≥2.01 ng/ml; GB: men with PSA ≥0.78 and <2.01 ng/ml; CG: men with PSA <0.78ng/ml. PSA velocity negative: the value of the PSA decreases over time; PSA velocity positive: the value of PSA increases over time.
20.-Page 8. Conclusion.
I do not agree with the conclusion. From the information provided, this study demonstrates that in patients who are interested in prostate health, patients with high PSA are more likely to undergo biopsy. In this group, PSA velocity greater than 0.75ng/ml was associated with prostate cancer diagnosis, although the baseline data is required to conclude this.
ANSWER:
The author is correct in his statement that the study can conclude:
in patients who are interested in prostate health, patients with high PSA are more likely to undergo biopsy. In this group, PSA velocity greater than 0.75ng/ml was associated with prostate cancer diagnosis
but our multivariate analysis shows that
PSA velocity, the direction of PSAV acceleration and the free PSA / total PSA index, at cut-off points of PSA 0.78ng/ml and 2.01ng/ml
discriminate benign from malignant pathology. The key to being able to state the conclusion comes from the highly rigorous multivariate analysis suitable for this type of individuals participating in the study.
WE CHANGE WHERE IT SAYS:
PSA velocity, the direction of PSAV acceleration and the free PSA / total PSA index, at cut-off points of PSA 0.78ng/ml and 2.01ng/ml in a priori healthy men over 47, predict the probability of benign or malign pathology of the prostate.
BY:
In a priori healthy men over 47who are interested in prostate health, patients with high PSA are more likely to undergo biopsy. In this group, PSA velocity greater than 0.75ng/ml is associated with prostate cancer diagnosis. PSA velocity, the direction of PSAV acceleration and the free PSA / total PSA index, at cut-off points of PSA 0.78ng/ml and 2.01ng/ml predict the probability of benign or malign pathology of the prostate.
21.-There would be extensive data from this cohort:
We add this paragraph and table 2 about this issue:
In the cohort of prostate cancer diagnosis (table 2), mean age was 67.30 years (SD 8.11), median 66.00, range 53-86; there was no differences (p = 1.000). Mean BMI was 26.97 kg / m2 (SD 3.35), median 27.00, range 20.00–36.00; there were no differences between groups (p = 0.762). Mean PSAV was 3.13, SD 5.41, median 1.33, range 0.001–34.46; there were no differences between groups (p = 0.828). Mean PSA index was 17.16 u / L (SD 8.85), median 15.03, range 5.03–48.53; PSA index was lowest in GA (p = 0.0041). PSAV had more positive acceleration in GA and more negative acceleration in GC (p = 0.0036). Mean Gleason was 7.30 (SD 0.755), median 7.00, range 5-8, was lowest in GC (p = 0.00004).
Table 2. -Main variables in men with prostate cancer diagnose in the groups.
|
Variable |
Group |
p |
|||||
|
GAP Ca+ |
GB PCa+ |
GC PCa+ |
|||||
|
|
Mean (SD) |
Median (range) |
Mean (SD) |
Median (range) |
Mean (SD) |
Median (range) |
|
|
Age |
67.59 (8.16) |
66.50 (53.00-86.00) |
62.40 (7.76) |
65.00 (55.00-72.00) |
66.33 (7.20) |
68.00 (53.00-74.00) |
1.0000 |
|
BMI |
26.85 (3.38) |
26.00 (20.00-36.00) |
29.30 (2.00) |
29.00 (27.00-31.00) |
27.33 (3.50) |
26.50 (23.00-32.00) |
0.762 |
|
PSAV |
3.21 (5.49) |
1.38 (0.001-34.46) |
0.20 (0.17) |
0.19 (0.07-0.32)) |
2.25 (0.20) |
2.20 (2.20-2.35) |
0.828 |
|
iPSA |
16.40 (8.39) |
14.76 (5.03-48.53) |
21.55 (13.87) |
16.35 (7.06-39.29) |
26.41 (6.67) |
26.39 (18.20-34.94) |
0.0041 |
|
PSAV acceleration |
Positive (n/%) |
Negative (n/%) |
Positive (n/%) |
Negative (n/%) |
Positive (n/%) |
Negative (n/%) |
0.0036 |
|
76/74.5 |
26/25.50 |
2/30.00 |
3/70.00 |
1/16.70 |
5/83.30 |
||
|
Gleason |
7.47 (0.50) |
7.00 (7-8) |
6.60 (0.54) |
7.00 (6-7) |
5.10 (0.10) |
5.00 (5-5.10) |
0.00004 |
and I believe with some modifications, this would be suitable for publication. We look forward to seeing the next version.
Reviewer 2 Report
The authors studied the PSA velocity and the free / total ratio in a population of 2035 men presumed to have no prostate problem.
They report the interest of PSA velocity and iPSA for predicting the probability of benign or malignant pathology of the prostate.
What was known so far is summarized in the article not cited in this publication: “PSA velocity is uninformative of risk at diagnosis; high PSA velocity is not an indication for treatment in men on active surveillance; PSA velocity at the time of recurrence should be entered into prediction models (or "nomograms") to aid patient counseling” (A commentary on PSA velocity and doubling time for clinical decisions in prostate cancer. Vickers et al. Urology . 2014 Mar;83(3):592-6).
The following reference cited in the article is also critical of the value of the PSA velocity
“PSAV calculation has been advocated by many investigators as a strategy to improve the screening and clinical management of patients with CaP. However, when PSAV definitions are rigorously applied, its calculation does not significantly enhance the clinical performance of PSA alone” (PSA velocity: a systematic review of clinical applications”. (Loughlin. Urol Oncol. 2014 Nov;32(8):1116-25).
Can the information provided by the authors change the practice? it is not certain
Changes to be made:
Figure 1 is not clear. Is ”negative and positive” PSAV acceleration ? the figure is not indexed in the text.
Reference 20 should be completed
Author Response
We respond to reviewer 2 in blue
Comments and Suggestions for Authors
The goal of this study was to determine the relationship between measures of PSA (value, acceleration, and ratio) and prostate conditions (BPH, PIN, prostatitis, and prostate cancer.) The study is descriptive and somewhat detailed. However, there are a number of concerns that should be addressed.
General issues
1.-The title does not reflect the goal of the study. Consider revising the title.
ANSWER:
The aim of the study is to establish the association between prostate-specific antigen velocity (PSAV), value and acceleration and of the free PSA/total PSA index or ratio with prostate conditions.
THEREFOR, WE CHANGE:
Prostate-specific antigen velocity (PSAV) as a positive predictive marker of prostate conditions.
BY:
The association between prostate-specific antigen velocity (PSAV), value and acceleration, and of the free PSA/total PSA index or ratio, with prostate conditions.
2.-Numerous studies have focused on this relationship. Novel aspects of this project should be emphasized.
ANSWER:
WE ADD THIS PARAGRAPH TO THE DISCUSSION:
Although other studies have focused on the relationship between PSA velocity and prostate cancer, our study provides important novelties. The first is that it has been carried out with a completely new design: the study has been carried out on men from the community, who when faced with a public announcement in a social communication medium, such as the newspaper in the written press, which covers all the population and has a homogeneous and complete diffusion throughout the geography of the Salamanca Health Area (La Gaceta de Salamanca). This is a fact that makes the inclusion of participating individuals original. Therefore, this investigation of PSA velocity and acceleration is very interesting. The second novelty of impact is that the direction of the acceleration of the PSA is taken into account: positive or negative, in the population group of the community. And the third novelty is that it is not a study on the relationship between PSAV and prostate cancer, but rather a study between PSAV and any prostate condition: BPH, PIN, non-infectious prostatitis or cancer. And this is completely new.
Abstract
3.-It is unclear how the group cut-points were decided. Please indicate the method used to determine the PSA range that defined each group.
ANSWER:
This paragraph is added to Methods:
To decide the study groups, two factors were taken into account:
-The value of PSA in all of the 2035 individuals: mean 2.01, SD 10.01, median 1.1, range 0.04-435.94.
-The distribution of the values: on one hand, it was tried that the groups had a number of individuals as similar as possible. On the other hand, that the value of the PSA figures could have a clinical significance.
4.-Indicate how prostate conditions were assessed (medical records, prostate tissue review, etc.)
ANSWER:
We add this paragraph to Methods:
The diagnosis of the prostate situation was made according to the ordinary practice of both primary care physicians and urology specialists: anamnesis, physical examination, routine complementary examinations of the study of prostate health.
5.-The statistical methods should be summarized.
ANSWER:
WE CHANGE:
The automatic statistics calculator NSSS2006/GESS2007 was used. Results were analysed with descriptive statistics, Student's t, Chi2, Fisher's exact test, ANOVA (with Scheffe's test for normal samples and Kruskal-Wallis for other distributions), and multivariate analysis. Statistical significance was accepted for p <0.05.
BY:
The automatic statistics calculator NSSS2006/GESS2007 was used. Results were analysed with descriptive statistics, Student's t, Chi2, Fisher's exact test, ANOVA (with Scheffe's test for normal samples and Kruskal-Wallis for other distributions), and multivariate analysis. For the multivariate analysis, two highly rigorous statistical techniques were used for this type of distribution, that is, to analyze individuals from the community with great differences, despite being all of the same race, culture, lifestyle and same Area of Health. First, a multidimensional scaling analysis was carried out with the non-metric PROXSCAL method, which allows spatial representation, in the form of a map, and shows the proximity between the variables and the analysis of the relationship between them. A stress analysis of the model itself was carried out to check its usefulness in this analysis and in this population, resulting in that the applied model was perfect and adjusted well to the variables in each dimension, with a normalized gross stress of 0.5%. The model showed that the number of dimensions to represent the variables was three, being the most appropriate setting for the representation. The model displays the variables and their distances (coordinates) within each dimension, which determines the representation of each variable in the dimensions. The other multivariate analysis method used was the vector model. The objective of this model is to represent the variables in the investigated spaces or dimensions: it shows the projection of the variables and the correlation between the values of each one and the projection in the dimensions between the groups. Statistical significance was accepted for p <0.05.
6.-Define negative acceleration.
ANSWER: We add this paragraph to Methods:
The PSA velocity is the value of the PSA change over time, adjusted to 12 months. It was called positive acceleration when the velocity was positive, that is, when the PSA value was increasing. It was called negative acceleration when the velocity was negative, that is, when the PSA value was decreasing.
7.-Indicate if the analyses were adjusted for age.
ANSWER: We add this paragraph to Methods:
This multivariate analysis includes an adjustment of the variables for age.
8.-Indicate which results were statistically significant.
ANSWER: We add the p value in multivariate análisis.
Introduction
9.-Present a study rationale. Also, the main concepts are disjointed. Please make the introduction a bit more cohesive.
ANSWER:
We organize the concepts and paragraphs of the introduction.
We begin the introduction with a paragraph that justifies the study: we add this paragraph:
Prostate health problems, both benign and malignant, control more than half of the entire male population (Salvatierra 2013, Padilla 2013). Although the main laboratory tool for its study is PSA, this is a non-specific and not very sensitive marker, therefore, we propose a study that provides information about the analysis of the velocity and direction of PSA velocity in the male population of the community in its relation to prostate health, both benign and malignant.
10.-Clarify the statement at lines 79-80.
ANSWER:
WE CHANGE:
Biopsies detected PCa in 56% of men with iPSA <0.10 and only in 8% of men with iPSA> 0.25 ng/mL, with a PSA of 4–10 ng/mL (Catalona, Partin et al. 1998).
BY:
When prostate biopsies are done in men with PSA between 4-10 ng / ml, in the case of having iPSA <0.10 the probability of being positive for cancer is 56%, and when iPSA> 0.25 they are positive for cancer alone the 8% (Catalona, Partin et al. 1998).
Methods
11.-Indicate which community was the recruitment site.
ANSWER:
We add in Methods:
The Health Area from which individuals from the community were recruited was the province of Salamanca (Spain).
12.-Provide more detail about the number and percent of patients biopsied.
ANSWER: it has been added.
Define negative and positive PSA velocity.
ANSWER: it has been added.
13.-As mentioned above, it is unclear how the group cut-points were decided. Please indicate the method used to determine the PSA range that defined each group.
ANSWER: it has been added.
Results
14.-Label the x-axis on Figure 1:
ANSWER: it has been done.
15.-The information provided at lines 172-196 seem extraneous. It is not clear how these characteristics relate to the goals of the study.
ANSWER:
We add this paragraph in discussion:
Although the objectives of the study are to know the relationship of changes in PSAV and iPSA with the main diagnosis of prostate status, it is convenient to take into account circumstances such as medication or concomitant diseases that could influence the results. Therefore, all diagnoses have been investigated, both of the urinary system and other parts of the body and the concomitant medications.
16.-Figures 2-3 are difficult to read. A larger font is needed. Perhaps display in a table instead. Also, indicate statistical significance.
ANSWER: We change the font in the figures.
Discussion
17.-Start with primary study findings. Indicate how study findings relate to findings of other studies (compare to the literature).
Answer: we change it.
18.-Correct CG… should be GC in the third paragraph and at line 229.
Answer: we change it.
19.-Explain how the finding of ranitidine inversely associated with PSA reflects the literature. Is there a mechanism that might explain this relationship?
ANSWER:
We add this paragraph in discussion:
Ranitidine, like famotidine, belongs to the group of histamine antagonist drugs at type II histamine receptors, also called H2. They are used in the treatment of gastrointestinal diseases such as gastric and duodenal ulcer, gastroesophageal reflux, and other pathological hypersecretory conditions (Taha, Hudson et al. 1996). One of the effects of H2 antagonists is hyperprolactinemia (Salazar-López-Ortiz, Hernández-Bueno et al. 2014). The relationship between testosterone level and prostate disease is controversial. While some authors claim that there is no association between prostate cancer and serum levels of testosterone and prostate antigen (Ruiz López, Pérez Mesa et al. 2017), other authors claim that testosterone is significantly lower in patients with prostate cancer than in controls (Rivera G., Tagle A. et al. 2003). Therefore, it is an active research area to find the relationship between testosterone levels and prostate disease.
20.-Provide the clinical significance of this study and ideas for future studies.
ANSWER:
We add this paragraph:
Our study in male individuals from the community demonstrates the importance of changes in PSA velocity and the influence of treatments that modify testosterone levels in a chronic and sustained way, such as ranitidine-type gastric protectors. These findings make us progress in these lines of research in the future.
Reviewer 3 Report
The goal of this study was to determine the relationship between measures of PSA (value, acceleration, and ratio) and prostate conditions (BPH, PIN, prostatitis, and prostate cancer.) The study is descriptive and somewhat detailed. However, there are a number of concerns that should be addressed.
General issues
- The title does not reflect the goal of the study. Consider revising the title.
- Numerous studies have focused on this relationship. Novel aspects of this project should be emphasized.
Abstract
- It is unclear how the group cut-points were decided. Please indicate the method used to determine the PSA range that defined each group.
- Indicate how prostate conditions were assessed (medical records, prostate tissue review, etc.) The statistical methods should be summarized.
- Define negative acceleration.
- Indicate if the analyses were adjusted for age.
- Indicate which results were statistically significant .
Introduction
- Present a study rationale. Also, the main concepts are disjointed. Please make the introduction a bit more cohesive.
- Clarify the statement at lines 79-80.
Methods
- Indicate which community was the recruitment site.
- Provide more detail about the number and percent of patients biopsied. Define negative and positive PSA velocity.
- As mentioned above, it is unclear how the group cut-points were decided. Please indicate the method used to determine the PSA range that defined each group.
Results
- Label the x-axis on Figure 1.
- The information provided at lines 172-196 seem extraneous. It is not clear how these characteristics relate to the goals of the study.
- Figures 2-3 are difficult to read. A larger font is needed. Perhaps display in a table instead. Also, indicate statistical significance.
Discussion
- Start with primary study findings. Indicate how study findings relate to findings of other studies (compare to the literature).
- Correct CG… should be GC in the third paragraph and at line 229.
- Explain how the finding of ranitidine inversely associated with PSA reflects the literature. Is there a mechanism that might explain this relationship?
Provide the clinical significance of this study and ideas for future studies.
Author Response
Reviewer #3
Responses to reviewer 3 are in purple.
The authors studied the PSA velocity and the free / total ratio in a population of 2035 men presumed to have no prostate problem.
They report the interest of PSA velocity and iPSA for predicting the probability of benign or malignant pathology of the prostate.
1.-What was known so far is summarized in the article not cited in this publication: “PSA velocity is uninformative of risk at diagnosis; high PSA velocity is not an indication for treatment in men on active surveillance; PSA velocity at the time of recurrence should be entered into prediction models (or "nomograms") to aid patient counseling” (A commentary on PSA velocity and doubling time for clinical decisions in prostate cancer. Vickers et al. Urology . 2014 Mar;83(3):592-6).
ANSWER:
We add this in Discusión, which includes the reference recommended by the reviewer:
Until now, PSAV had only been investigated to determine whether it was related to the risk of prostate cancer diagnosis, the indication for treatment in men under active surveillance, or the ability to predict recurrence of prostate cancer (Vickers, Thompson et al. 2014). In our study, this is the first time that PSAV has been used to investigate any type of prostate pathology, including benign ones.
2.-The following reference cited in the article is also critical of the value of the PSA velocity
“PSAV calculation has been advocated by many investigators as a strategy to improve the screening and clinical management of patients with CaP. However, when PSAV definitions are rigorously applied, its calculation does not significantly enhance the clinical performance of PSA alone” (PSA velocity: a systematic review of clinical applications”. (Loughlin. Urol Oncol. 2014 Nov;32(8):1116-25).
ANSWER: we add this paragraph in Discussion:
Indeed, our study reports not only on the relationship of psav with prostate cancer that other authors have investigated (Loughlin 2014). Our research shows a relationship between changes in PSAV and both benign and malignant pathology, therefore, providing new scientific information.
3.-Can the information provided by the authors change the practice? it is not certain
ANSWER:
We add this paragraph in Discussion:
We believe that our study can modify routine determination habits of iPSA and PSAV. So far the iPSA has not been determined in our environment if the PSA is less than 4ng / ml. Our study shows that it is convenient to determine the iPSA at PSA values ​​lower than 4ng / ml since iPSA provides relevant clinical information on PSA lower than 4ng / ml. Regarding the velocity of PSA, it also advises changes in the usual clinical practice of the management of prostate disease, since in our study we observed that changes in PSAV, increasing or decreasing, provide information on whether the prostate disease is benign or malignant .
4.-Changes to be made:
Figure 1 is not clear. Is ”negative and positive” PSAV acceleration ?
ANSWER: We add this paragraph in thetext at the bottom of Figure 1: PSA velocity negative: the value of the PSA decreases over time; PSA velocity positive: increases the value of PSA over time.
5.- the figure is not indexed in the text.
ANSWER: we index the figure in te text.
6.-Reference 20 should be completed
ANSWER: we change and correct the reference
References
Catalona, W. J., A. W. Partin, K. M. Slawin, M. K. Brawer, R. C. Flanigan, A. Patel, J. P. Richie, J. B. deKernion, P. C. Walsh, P. T. Scardino, P. H. Lange, E. N. Subong, R. E. Parson, G. H. Gasior, K. G. Loveland and P. C. Southwick (1998). "Use of the percentage of free prostate-specific antigen to enhance differentiation of prostate cancer from benign prostatic disease: a prospective multicenter clinical trial." Jama279(19): 1542-1547.
Loughlin, K. R. (2014). "PSA velocity: a systematic review of clinical applications." Urol Oncol32(8): 1116-1125.
Rivera G., P., R. Tagle A., S. Mir C. and R. González I. (2003). "Relación entre niveles de tertosterona en suero y cáncer prostático." Actas Urológicas Españolas27: 788-792.
Ruiz López, A. I., J. C. Pérez Mesa, Y. Borrego Chi and Y. Cruz Batista (2017). "Testosterona y antígeno prostático específico en pacientes portadores de carcinoma prostático, provincia Holguín, 2013-2015." Correo Científico Médico21: 720-733.
Salazar-López-Ortiz, C.-G., J.-A. Hernández-Bueno, D. González-Bárcena, M. López-Gamboa, A. Ortiz-Plata, H.-L. Porias-Cuéllar, J.-D. Rembao-Bojórquez, G.-A. Sandoval-Huerta, R. Tapia-Serrano, G.-G. Vazquez-Castillo and V.-S. Vital-Reyes (2014). "Guia de práctica clínica para el diagnóstico y tratamiento de la hiperprolactinemia. ." Ginecol Obstet Mex82: 123-142.
Taha, A. S., N. Hudson, C. J. Hawkey, A. J. Swannell, P. N. Trye, J. Cottrell, S. G. Mann, T. J. Simon, R. D. Sturrock and R. I. Russell (1996). "Famotidine for the prevention of gastric and duodenal ulcers caused by nonsteroidal antiinflammatory drugs." N Engl J Med334(22): 1435-1439.
Vickers, A. J., I. M. Thompson, E. Klein, P. R. Carroll and P. T. Scardino (2014). "A commentary on PSA velocity and doubling time for clinical decisions in prostate cancer." Urology83(3): 592-596.
Round 2
Reviewer 1 Report
Nil further to comment.
Reviewer 3 Report
This is an improved version of the manuscript.
This manuscript is a resubmission of an earlier submission. The following is a list of the peer review reports and author responses from that submission.